# LP.8.1-directed COVID-19 mRNA vaccines durably boost neutralizing antibodies and mitigate ancestral immune imprinting

Ian A. Mellis[1,2☯*], Madeline Wu[1☯], Hsiang Hong[1,3☯], Anthony Bowen[1,4], Kristin Daniel[1,2], Carmen Gherasim[5], Virginia M. Pierce[5], Michael T. Yin[1,4], Aubree Gordon[6*], Yicheng Guo[1*], David D. Ho [1,4,7,8‡*]

1 Aaron Diamond AIDS Research Center, Columbia University Vagelos College of Physicians and Surgeons, New York, New York, United States of America, 2 Department of Pathology and Cell Biology, Columbia University Vagelos College of Physicians and Surgeons, New York, New York, United States of America, 3 Department of Microbiology, National Taiwan University College of Medicine, Taipei, Taiwan, 4 Division of Infectious Diseases, Department of Medicine, Columbia University Vagelos College of Physicians and Surgeons, New York, New York, United States of America, 5 Department of Pathology, University of Michigan, Ann Arbor, Michigan, United States of America, 6 Department of Epidemiology, University of Michigan, Ann Arbor, Michigan, United States of America, 7 Department of Microbiology and Immunology, Columbia University Vagelos College of Physicians and Surgeons, New York, New York, United States of America, 8 Pandemic Research Alliance unit at the Wu Center for Pandemic Research, Columbia University Vagelos College of Physicians and Surgeons, New York, New York, United States of America

☯ These authors contributed equally to this work.
‡ Lead contact.
* im2613@cumc.columbia.edu (IAM); gordonal@umich.edu (AG); yg2521@cumc.columbia.edu (YG); dh2994@cumc.columbia.edu (DDH)

## Abstract

As SARS-CoV-2 evolves, it evades existing immunity elicited by exposure to earlier strains of the virus. In response, vaccine manufacturers have updated COVID-19 vaccines annually since 2022, though immune imprinting to the ancestral strain has blunted antibody responses to modern viral variants. In early 2025, the JN.1 subvariant LP.8.1 was dominant and manufacturers updated mRNA vaccine formulations to target LP.8.1 (LP.8.1 MV). However, by late 2025, other subvariants were dominant (XFG and NB.1.8.1) or emerging (e.g., PE.1.4, BA.3.2, PY.1.1.1) around the world. It is critical to understand the extent to which updated vaccine boosters elicit titers against both their target strain and recent variants. Further, it is important to quantify the extent to which immune imprinting continues to shape antiviral immune responses. Using pseudoviruses, we measured neutralizing antibody titers against a panel of 11 SARS-CoV-2 variants in serum samples from 36 adult participants in the United States before and approximately 1 month after LP.8.1 MV booster. We found that neutralizing antibody titers were substantially increased by the boost, with the greatest increases elicited against LP.8.1 and XFG. For the first time, post-boost titers were higher against the homologous vaccine target (LP.8.1) than against D614G (representing the ancestral strain). Combined, these results indicate that ancestral

**Data availability statement:** De-identified raw neutralization data in spreadsheets with tables of contents are provided in a zip archive supplementary file.

**Funding:** This study was supported by funding from the NIH SARS-CoV-2 Assessment of Viral Evolution (SAVE) Program (subcontract no. 0258-A700-4609 under federal contract no. 75N93021C00014) to D.D.H. and (subcontract GR0010139-PO024016 under federal contract no. 75N93021C00016) to A.G., the Gates Foundation (project INV019355) to D.D.H., internal startup funding UR014016 from Columbia University to Y.G., K24 AI155230 to M.T.Y., and K08 AI180347 to A.B. NIH: https://www.nih.gov/ Gates Foundation: https://www.gatesfoundation.org/ Columbia University: https://www.columbia.edu/ The funders did not play any role in the study design, data collection and analysis, decision to publish, or preparation of the manuscript.

**Competing interests:** I have read the journal's policy and the authors of this manuscript have the following competing interests: David D. Ho co-founded TaiMed Biologics and RenBio, and he serves as a consultant for Brii Biosciences and is a board director at Vicarious Surgical. Aubree Gordon served as a member of the scientific advisory board for Janssen Pharmaceuticals and has consulted and serves on a scientific advisory board for Sanofi Pasteur. The remaining authors declare no conflicts of interest.

immune imprinting is mitigated to the greatest extent observed to date by LP.8.1 MV. Lastly, for a subset of participants, we measured neutralizing titers at approximately 4 months post-booster and found that LP.8.1-directed antibody titers were durable, with an estimated average half-life of approximately 66 days.

---

## Author summary

SARS-CoV-2, the virus that causes COVID-19, has evolved since it first emerged in 2019, demonstrating progressive evasion of existing antiviral immunity in the human population. In response, vaccine manufacturers have updated COVID-19 vaccines annually since 2022 to target the most recent dominant immune-evasive viral variants. The 2025–2026 formulations of mRNA vaccines in the U.S., Europe, Japan, and other countries target the LP.8.1 variant (LP.8.1 MV boosters), but other strains, such as XFG and NB.1.8.1 have become dominant since these boosters were developed, and yet more distant strains, such as BA.3.2, have begun to spread around the world, as well. It is essential to understand the potency and durability of vaccine-elicited antibody responses against evolving viral variants, and to assess whether prior impediments to effective vaccine responses, such as immune imprinting, persist. By analyzing blood samples from longitudinally sampled human participants who received LP.8.1 MV boosters, we show that LP.8.1 MV boosters elicit potent and durable immune responses against the dominant SARS-CoV-2 variants of concern. We then consistently reanalyze several historical booster-elicited immune response datasets generated after prior vaccine formulations and find that LP.8.1 MV boosters elicit the strongest target-directed responses to date, dramatically overcoming prior limitations, such as immune imprinting.

## Introduction

As SARS-CoV-2 evolves, it evades existing immunity elicited by exposure to earlier strains of the virus. In response, vaccine manufacturers have updated COVID-19 vaccines annually since 2022, though immune imprinting to the ancestral strain has blunted antibody responses to modern variants [1–3]. Throughout 2025, the spread of SARS-CoV-2 has been dominated by a series of subvariants in the JN.1 sublineage, with intermittent concern for divergent lineages, such as BA.3.2 [4,5]. In early 2025, the JN.1 subvariant LP.8.1 was dominant. LP.8.1 is more resistant to serum antibody neutralization than early JN.1 subvariants [4]. In August 2025, the U.S. Food and Drug Administration authorized two updated mRNA vaccines (Pfizer-BioNTech and Moderna) targeting LP.8.1. In the U.K., E.U., and Japan, LP.8.1-based mRNA vaccines were also authorized. Here we report the boosting effect of updated LP.8.1 monovalent mRNA vaccines (LP.8.1 MV) on serum SARS-CoV-2-neutralizing antibodies in healthy adults in the United States. Others have reported peak boosting

effects in other populations in Germany and Japan [6,7]. However, it remains essential to characterize boosting effects directly in the U.S. population, due to its different SARS-CoV-2 variant exposure, immunization, and COVID-19 incidence history [4,8,9]. Critically, because recent shifts in U.S. COVID-19 vaccination policy have created substantial uncertainty regarding age-based booster recommendations, we assess boosting effects in adults across multiple age ranges, including those younger than 50-years-old, 50-to-64-years-old, and 65-years-or-older. Lastly, it is important to characterize the durability of LP.8.1 MV boosting effects, which has not yet been discussed by others.

Since authorization of LP.8.1 MV boosters, SARS-CoV-2 has evolved beyond LP.8.1, with JN.1 subvariants XFG and NB.1.8.1 becoming dominant globally. Other JN.1 subvariants, such as PY.1.1.1 in North America, PE.1.4 in Oceania, and the highly divergent BA.3.2 sublineage, mostly in Australia and Germany, are also growing in incidence (S1 Fig). XFG, NB.1.8.1, PE.1.4, and PY.1.1.1 contain 8, 5, 10, and 10 spike amino-acid differences relative to LP.8.1, respectively, and BA.3.2 has dozens of amino-acid differences, in a separate viral sublineage entirely (Fig 1A, S2 Fig). It is critical to assess the effectiveness of the updated LP.8.1 MV boosters in human sera against recently dominant subvariants and those that are of growing concern. Furthermore, it is important to understand the extent to which ancestral immune imprinting still shapes humoral immunity against circulating SARS-CoV-2 variants.

## Results and discussion

To assess serum neutralizing antibodies elicited by LP.8.1 MV boosters, we generated vesicular stomatitis virus (VSV)-based pseudoviruses for historic and circulating SARS-CoV-2 variants: D614G, BA.5, XBB.1.5, JN.1, KP.2, LP.8.1, NB.1.8.1, XFG, PE.1.4, BA.3.2, and PY.1.1.1. We then performed pseudovirus neutralization assays using serum samples collected before and approximately 1 month (mean 28.6 days, range 21–45 days) after vaccination (S1, S2 Tables). Vaccination boosted serum virus-neutralizing 50% inhibitory dilution ($ID_{50}$) titers by 1.9-fold against D614G, 2.5-fold against BA.5, and 3.1-fold against XBB.1.5 (Fig 1B), reflecting persistent but modest back-boosting comparable to last year's KP.2 booster. However, LP.8.1 MV induced a more substantial boost (5.2- to 10.6-fold) against subvariants in the JN.1 lineage, with the greatest increases against LP.8.1 (10.3-fold) and the present dominant strain XFG (10.6-fold). Pre-boost titers (Geometric Mean $ID_{50}$ [GMT]), reflective of much of the U.S. population, varied widely across variants (350–2,259), with highest titers against D614G and lowest against PY.1.1.1. These pre-boost titers suggest that a substantial fraction of the population may have low baseline protection from infection by variants such as BA.3.2 (GMT = 505) and XFG (GMT = 362), based on our prior studies of neutralizing antibody correlates of protection [10]. However, post-boost virus-neutralizing titers were robust (1,317–9,065), with the highest value observed for LP.8.1 (GMT 9,065), correlated with greater protection in earlier studies. LP.8.1 MV also induced a modest boost (2.6-fold) against the highly divergent BA.3.2 (post-boost GMT 1,317), though post-boost titers against BA.3.2 were lower than post-boost titers against other tested variants. This marks the first time that we have observed post-boost titers higher against the homologous target strain than against the ancestral D614G[1–3].

To assess antigenicity of the tested variants, we performed antigenic mapping using the pre-boost and post-boost sample sets separately (Fig 1C). The relative positions of variants in the antigenic map were largely preserved before and after boosting, suggesting that neutralization correlation patterns across variants did not substantially change. However, pre-boost sera were embedded primarily near D614G and BA.5 whereas post-boost sera shifted toward LP.8.1, concordant with maximum post-boost titers against LP.8.1. Combined with dramatic homologous LP.8.1-boosting and low back-boosting, these results suggest that LP.8.1 MV vaccination substantially mitigated ancestral immune imprinting. This mitigation of imprinting is associated both with increased genetic distance of LP.8.1 from the imprinted ancestral strain and with an accumulated larger number of monovalent Omicron subvariant exposures in this cohort compared with earlier studies. For example, in this cohort the average total number of monovalent Omicron-directed vaccine doses (i.e., XBB.1.5 MV, KP.2 MV, and LP.8.1 MV combined) was 2.6, while in our prior study of KP.2 MV boosters it was 1.8 [3].

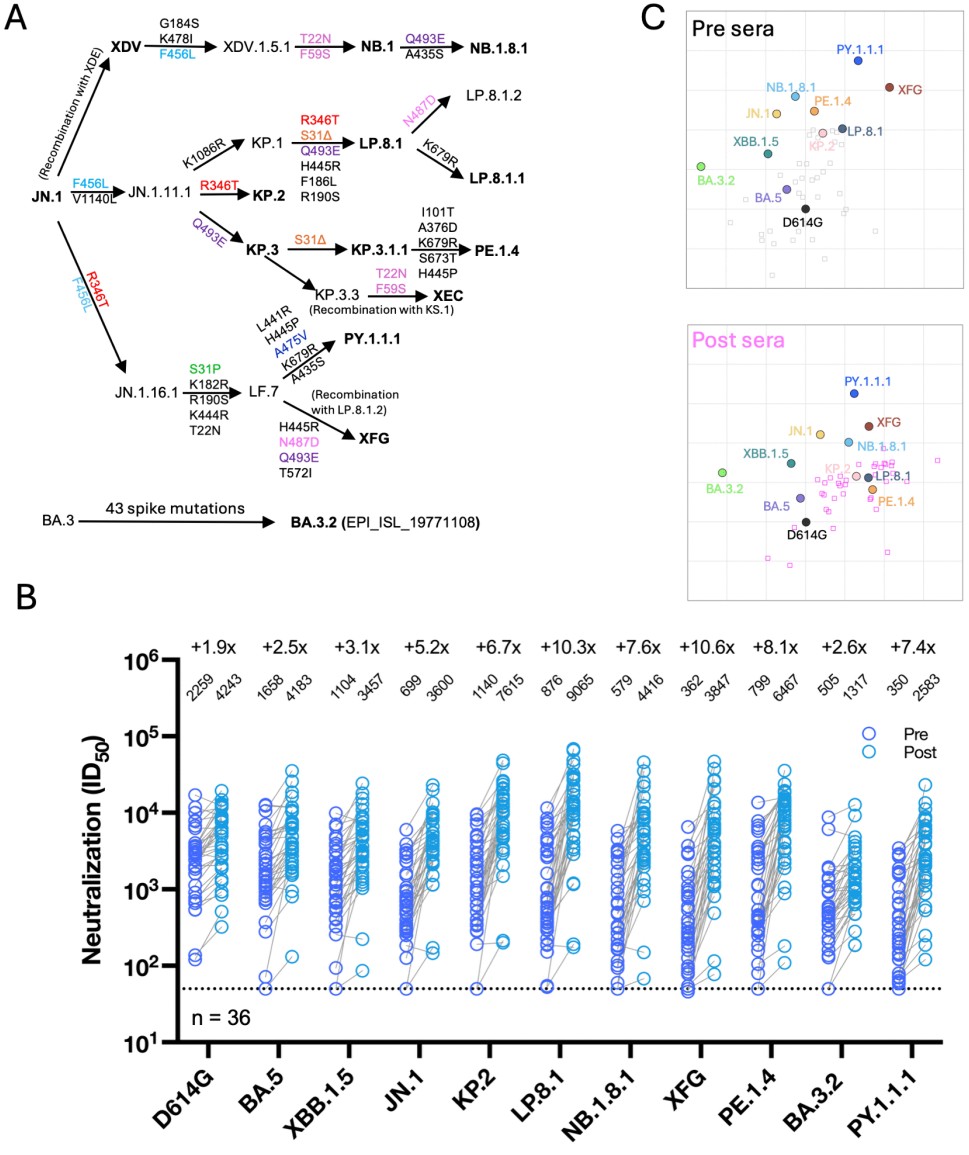

**Fig 1. SARS-CoV-2-neutralizing antibody titers before and after a LP. 8.1 monovalent mRNA vaccine booster. A.** Spike mutations of the indicated JN.1 sublineages. Δ, deletion. **B.** Serum virus-neutralizing titers ($ID_{50}$) against the indicated SARS-CoV-2 pseudoviruses and the fold changes between pre- and post-booster serum samples are shown. Geometric mean titers (GMT) are shown above each sample set, and the fold change from pre- to post-booster is shown above GMTs. MV, monovalent vaccine. n, sample size. The dotted line represents the assay limit of detection (LOD) of 50. **C.** Antigenic mapping of SARS-CoV-2 variants before and after a LP.8.1 monovalent mRNA vaccine booster.

To quantify the effects of immune imprinting on neutralizing antibodies elicited by different boosters, we compared LP.8.1 MV booster-elicited titers to titers we previously measured in earlier cohorts boosted with BA.5 BV, XBB.1.5 MV, or KP.2 MV [1–3]. First, we compared the fold-changes in titers against the imprinted D614G strain and the titers against the target strain of each vaccine (i.e., BA.5 for BA.5 BV, XBB.1.5 for XBB.1.5 MV, and KP.2 for KP.2 MV). Consistent with prior results, we found that monovalent Omicron-subvariant-directed vaccines elicited larger target fold changes (7.8X-13X) than ancestral-directed titer back-boosts (1.6-3.2X), particularly when compared to the BA.5 BV formulation (2.6X

target-directed, 1.9X ancestral-directed; Fig 2A). Next, we compared post-boost titers against the target strains of each of the four boosters versus post-boost titers against D614G. We found substantially larger fold differences in post-boost titers after LP.8.1 MV (2.1X, LP.8.1 vs. D614G) compared to earlier boosters (0.09X-0.22X, target vs. D614G), suggesting that the strength of LP.8.1 MV booster-elicited target-directed responses is not limited by immune imprinting as much as in earlier years (Fig 2B).

To better understand how LP.8.1 MV boosting performs across the adult age spectrum under current age-focused vaccination policies, we stratified the 36 participants into three age groups. 18 participants were between 18–49-years-old, 10 were 50–64-years-old, and 8 were 65-years-old or older (full cohort mean 47.3, range 19–80). All age groups demonstrated similar trends in boosting: the fold-increases against D614G were 1.9X, 1.9X, and 1.8X, respectively, and fold-increases against LP.8.1 were 9.7X, 8.9X, and 14.4X (S3 Fig). Post-boost titers were potent and broad cohort-wide, but

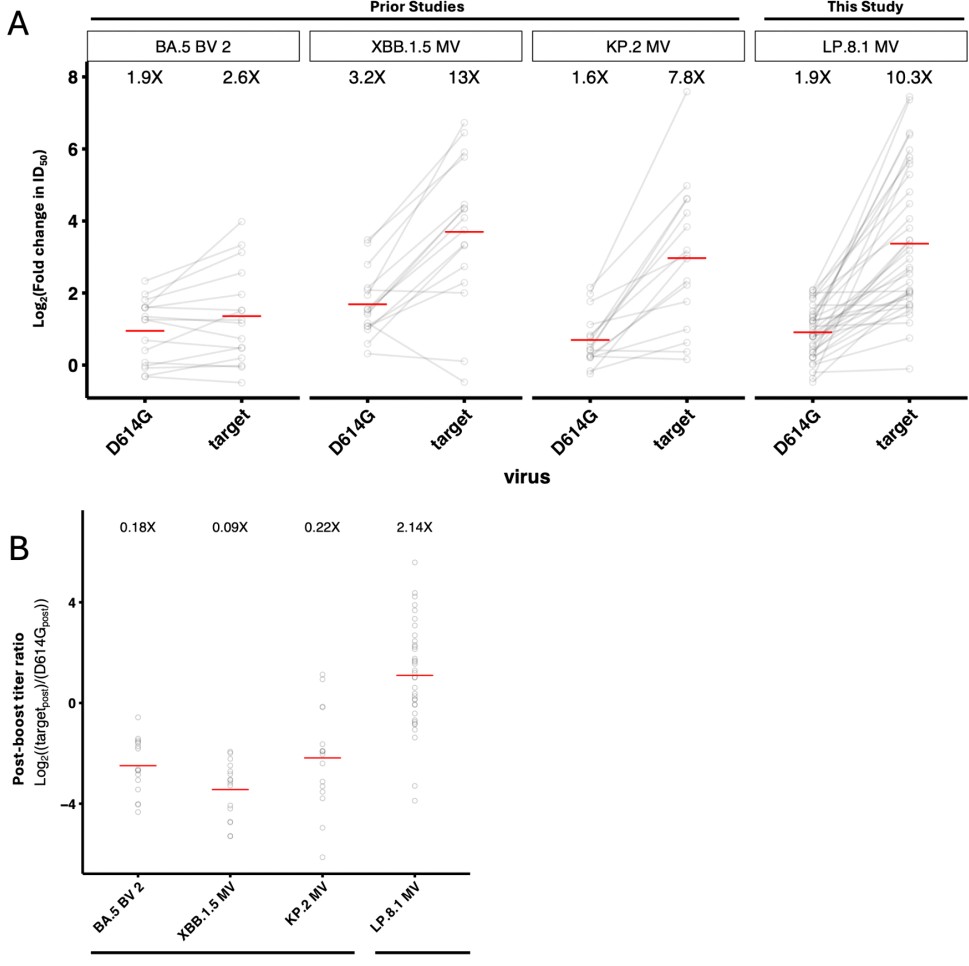

**Fig 2. Mitigation of immune imprinting by Omicron-subvariant-directed monovalent mRNA vaccines. A.** Fold change in ID50 against D614G or the target variant of each vaccine, as measured in pseudovirus neutralization assays. BA.5 BV 2, second bivalent wildtype/BA.5 bivalent booster; XBB.1.5 MV, XBB.1.5 monovalent vaccine booster; and KP.2 MV, KP.2 monovalent vaccine booster data from previously published studies (citations). Red lines indicate geometric mean fold change in titer, which is noted above each group of results. **B.** Fold differences in post-boost ID50 against the target variant of each vaccine vs. against D614G, using the same data sources (target$_{post}$/ D614G$_{post}$). Red lines indicate geometric mean fold difference, which is also noted above each group of results.

titers against JN.1 subvariants were highest in the oldest age group. Post-boost GMTs against D614G were 4,810, 3,947, and 3,504, respectively, while those against LP.8.1 were 6,795, 7,551, and 21,787. These levels of neutralizing antibodies were predictive of protection against symptomatic COVID-19 in prior clinical studies, but the precise titers that correlate with protection will vary according to the assays employed. Importantly, in addition to different numbers of participants in each age group, there were differences in numbers of reported lifetime COVID-19 vaccine doses: 5.9, 7, or 8 total doses respectively, of which an average of 3, 3.4, or 3.6 were the wild-type formulation, 0.6, 1, or 0.9 were wild-type/BA.5 bivalent, and 2.3, 2.6, or 3.5 were monovalent Omicron-subvariant-directed vaccines (S1, S2 Tables). Therefore, here post-boost titer differences between age groups may be confounded with vaccination history.

To assess the durability of antibody responses to LP.8.1 MV boosters, we measured neutralizing titers against a panel of 8 SARS-CoV-2 variants at approximately 1 and 4 months post-boost for 11 participants (S3, S4 Tables). We found that titers were durable over the study period in this cohort, with half-lives per variant ranging from >43 days against XFG and >66 days against LP.8.1 to >292 days against D614G (Fig 3). It is intriguing to note potential differences in variant-specific titer decay rates, with cross-reactive imprinting-associated titers, e.g., for D614G and perhaps also cross-reactive to BA.3.2, showing slower decay than newly elicited target-variant-specific titers.

In summary, LP.8.1 MV boosters elicit robust and durable neutralizing antibody responses against LP.8.1 and other JN.1 subvariants in healthy U.S. adults across age ranges. In this cohort, immune imprinting to the ancestral strain of SARS-CoV-2 was noticeably mitigated irrespective of age. Notably, and in contrast to prior reports in populations with different prior exposure histories, we observe highest post-boost titers against the homologous strain, LP.8.1 [6,7,9]. For example, in a study of non-immunosuppressed adults in the U.S., Lasrado and colleagues found that peak post-boost neutralizing titers against LP.8.1.1 were approximately half as high as against the tested ancestral strain WA1/2020 [9]. However, like in the present study, they similarly saw a higher pre-to-post-boost titer increase against the target strain than against the ancestral,

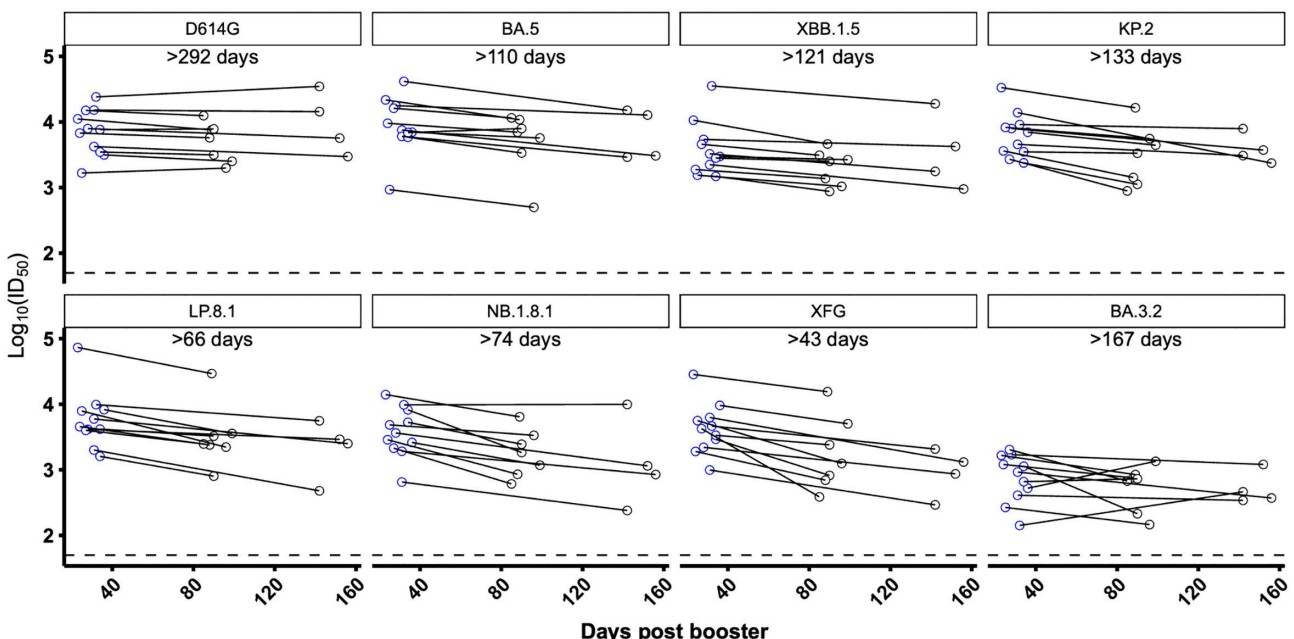

**Fig 3. Durability of neutralizing antibody titers after LP. 8.1 MV booster.** Serum virus-neutralizing titers ($ID_{50}$) for 11 participants against the 8 indicated SARS-CoV-2 pseudoviruses. Estimated half-life shown for each variant-directed titer. The dotted line represents the assay limit of detection (LOD) of 50.

with a 3.9X increase against LP.8.1.1 versus a 1.9X increase against WA1/2020, suggestive of mitigation of immune imprinting. Furthermore, we show for the first time that LP.8.1 MVs boost titers against other emerging variants of concern, including PE.1.4 and PY.1.1.1, and we characterize durability of LP.8.1 MV-elicited neutralizing antibody titers for the first time. The durability of titers against circulating and historical variants should continue to be monitored over the coming months, as should the clinical effectiveness of the vaccines in protecting from symptomatic infections and severe disease. Together, these results highlight how updating COVID-19 vaccines to combat contemporary viral variants is a promising and important strategy for mitigating suboptimal ancestral-strain-focused immune responses elicited by earlier exposures.

## Materials and methods

### Ethics statement

Serum samples were collected through the VIVA study at the University of Michigan and through the C-PIC study at Columbia University. Specimens were obtained following formal written informed consent in adherence to the protocols approved by the Institutional Review Boards of the University of Michigan Medical School (IRBMED) (protocol HUM00232359) and Columbia University IRB 1 (protocol AAAS9722).

### Clinical cohort sampling

Serum samples were collected from individuals who had received the LP.8.1 monovalent vaccine booster (LP.8.1 MV). Samples were collected before and after booster administration. Most of the study subjects were female (77.8%) with an average age of 47.3 years. Pre-, approximately-1-month-post-, and approximately-4-month-post-boost samples were examined by anti-nucleoprotein (NP) ELISA to check for unreported SARS-CoV-2 infections. Further details, vaccination histories, and sampling timelines are in S1, S2, S3, and S4 Tables.

### Cell lines

HEK293T (ATCC, CRL-3216) cells and Vero-E6 cells (ATCC, CRL-1586) were cultured in Dulbecco's Modified Eagle's Medium (DMEM) plus 10% heat-inactivated fetal bovine serum (FBS) and 1% penicillin-streptomycin (PS) in an atmosphere of 5% $CO_2$ at 37°C.

### SARS-CoV-2 spike plasmids

The spike constructs of D614G, BA.5, XBB.1.5, JN.1, KP.2, LP.8.1, XFG, and NB.1.8.1 were previously generated [3,4]. Constructs for PE.1.4, BA.3.2, and PY.1.1.1 were generated using previously reported methods [11]. All constructs were confirmed by whole-plasmid sequencing prior to packaging.

### SARS-CoV-2 pseudovirus packaging

VSV-based SARS-CoV-2 variant pseudoviruses were produced per previously established protocol [12].

### Pseudovirus neutralization

Pseudoviruses were titrated prior to each neutralization assay, and assays were performed per previously established protocol [4]. The serum dilution that inhibits 50% of virus entry ($ID_{50}$) was calculated using five-parameter dose-response curve fitting using R package drda_v2.0.3 in R_v4.3.2 [13].

### Quantification and statistical analysis

$ID_{50}$ values at or below the assay limit of detection (LOD) of 50 were treated as 50 for the purpose of calculating each group's geometric mean $ID_{50}$ titer (GMT). Half-lives were calculated using an exponential model. For individuals with no

change or positive change in titer between 1- and 4-months samples, half-life of that titer was treated as 365 days for use in calculation of average half-life.

## Supporting information

**S1 Table. Summary of clinical cohort for pre- and post-boost analysis.** No., number; y.o., years old; WT, wildtype; MV, monovalent vaccine; BV, bivalent vaccine.
(DOCX)

**S2 Table. Demographics of clinical cohort for pre- and post-boost analysis.** Vaccine formulations are denoted as wildtype (WT), BA.5 Bivalent (BA.5), XBB.1.5 monovalent (XBB.1.5), and KP.2 monovalent (KP.2). Vaccine manufacturers are denoted as Pfizer (P), Moderna (M), and Unknown (U). Yr, years; Infx, infection; Vax, vaccination; DBV, days before vaccination; DPV, days post vaccination; DPI, days post infection; F, female; M, male; Wh, white; Af, Black or African American; As, Asian.
(DOCX)

**S3 Table. Summary of clinical cohort for durability post-boost analysis.** No., number; y.o., years old; WT, wildtype; MV, monovalent vaccine; BV, bivalent vaccine; 1m, ~1 month post-boost sample; 4m, ~4 month post-boost sample.
(DOCX)

**S4 Table. Demographics of clinical cohort for durability post-boost analysis.** Vaccine formulations are denoted as wild-type (WT), BA.5 Bivalent (BA.5), XBB.1.5 monovalent (XBB.1.5), and KP.2 monovalent (KP.2). Vaccine manufacturers are denoted as Pfizer (P), Moderna (M), and Unknown (U). Yr, years; Infx, infection; Vax, vaccination; DBV, days before vaccination; DPV, days post vaccination; DPI, days post infection; F, female; M, male; Wh, white; Af, Black or African American; As, Asian.
(DOCX)

**S1 Fig. Frequencies of SARS-CoV-2 variants around the world.** Relative frequencies of SARS-CoV-2 variants from 08/01/2025–02/01/2026 in the indicated regions. Numbers of sequences analyzed in parentheses above each subpanel. Data from GISAID.
(TIF)

**S2 Fig. Spike amino acid differences between selected JN.1 subvariants.** Spike amino acid differences between JN.1 sub-variants of interest. BA.3.2 is in the BA.3 lineage, genetically distant from JN.1, and is not included in this table. See Fig. 1A.
(TIF)

**S3 Fig. LP.8.1 MV boost effects by KP.2 MV between age groups.** Data are presented as fold changes in neutralization $ID_{50}$ titers following LP.8.1 MV vaccination for across age groups. Geometric mean titers (GMT) are shown above each sample set, and the fold change from pre- to post-booster is shown above GMTs. MV, monovalent vaccine. n, sample size. The dotted line represents the assay limit of detection (LOD) of 50.
(TIF)

**S1 Data. Compressed zip file archive, which contains two files with all raw luminescence readings corresponding to presented neutralization results.** "Fig1-2_data.xlsx" contains data presented in Figs 1 and 2. "Fig3_data.xlsx" contains data presented in Fig 3. Each file contains a table of contents describing the component data tables.
(ZIP)

## Acknowledgments

We express our gratitude to Hiroshi Mohri (Columbia) for assistance with sample collection, to Jayesh G. Shah, Lawrence J. Purpura, Amanda Castillo, Meredith McNairy and Antonia Sturiza for conducting the C-PIC study (Columbia), and to Theresa Kowalski-Dobson, Anna Buswinka, Joseph Wendzinski, Mayurika Patel, Noah Paalanen and Ethan Hall of the VIVA study team for conducting the VIVA study.

## Author contributions

**Conceptualization:** Ian A Mellis, Aubree Gordon, Yicheng Guo, David D Ho.

**Data curation:** Ian A Mellis, Madeline Wu, Hsiang Hong, Yicheng Guo.

**Formal analysis:** Ian A Mellis, Madeline Wu, Hsiang Hong, Yicheng Guo.

**Funding acquisition:** Anthony Bowen, Michael T Yin, Aubree Gordon, Yicheng Guo, David D Ho.

**Investigation:** Ian A Mellis, Madeline Wu, Hsiang Hong, Anthony Bowen, Kristin Daniel.

**Methodology:** Ian A Mellis, David D Ho.

**Project administration:** Ian A Mellis, Madeline Wu, Hsiang Hong, Aubree Gordon, David D Ho.

**Resources:** Anthony Bowen, Carmen Gherasim, Virginia M Pierce, Michael T Yin, Aubree Gordon.

**Software:** Ian A Mellis.

**Supervision:** Ian A Mellis, Yicheng Guo.

**Validation:** Ian A Mellis, Madeline Wu, Hsiang Hong, Kristin Daniel, Yicheng Guo.

**Visualization:** Ian A Mellis, Madeline Wu, Yicheng Guo, David D Ho.

**Writing – original draft:** Ian A Mellis, Madeline Wu, Hsiang Hong, Yicheng Guo, David D Ho.

**Writing – review & editing:** Ian A Mellis, Madeline Wu, Hsiang Hong, Aubree Gordon, Yicheng Guo, David D Ho.

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
