## [Decision Letter · Decision Letter 0]

23 Apr 2026

PPATHOGENS-D-26-00788

LP.8.1-directed COVID-19 mRNA vaccines durably boost neutralizing antibodies and mitigate ancestral immune imprinting

PLOS Pathogens

Dear Dr. Ho,

Thank you for submitting your manuscript to PLOS Pathogens. After careful consideration, we feel that it has merit but does not fully meet PLOS Pathogens's publication criteria as it currently stands. Therefore, we invite you to submit a revised version of the manuscript that addresses the points raised during the review process.

We look forward to receiving your revised manuscript.

Kind regards,

Jesse D Bloom, Ph.D.

Guest Editor

PLOS Pathogens

Alexander Gorbalenya

Section Editor

PLOS Pathogens

Sumita Bhaduri-McIntosh

Editor-in-Chief

PLOS Pathogens

orcid.org/0000-0003-2946-9497

Michael Malim

Editor-in-Chief

PLOS Pathogens

orcid.org/0000-0002-7699-2064

**Additional Editor Comments :**

Thanks for submitting this valuable study to PLoS Pathogens. One of the reviewers has some constructive minor suggestions. Please incorporate those in a minor revision. I also suggest that you might add a citation and brief comparison/contrast of your results to this recent study in the Discussion (https://www.thelancet.com/action/showPdf?pii=S1473-3099%2826%2900050-2).

I expect I should be able to rapidly editorially review these minor revisions, so will not need to send the paper for re-review. But do include a brief reviewer response explaining any updates.

**Journal Requirements:**

4) Please ensure that the funders and grant numbers match between the Financial Disclosure field and the Funding Information tab in your submission form. Note that the funders and grants must be provided in the same order in both places as well.

**Reviewers' Comments:**

Reviewer's Responses to Questions

**Part I - Summary**

Reviewer #1: Mellis et al study presents data showing changes in neutralizing titers against different SARS-CoV-2 variants post LP.8.1 booster. They find that there is a broad increase in neutralizing titers post LP.8.1 booster across different SARS-CoV-2 variants, but notably the neutralizing titers are greatest against LP.8.1 and related SARS-CoV-2 variants. This is notable because in prior studies using earlier booster strains neutralizing titers post boost were always highest against D614G variant, which suggest that LP.8.1 booster vaccine might start overcoming immune imprinting caused by the early SARS-CoV-2 variant exposure.

Reviewer #2: The manuscript by Mellis et al is an excellent and important short report about the outcomes of LP.8.1 booster vaccines. A key point is that the booster is finally providing mitigating immune imprinting and providing broad protection. I have several cosmetic corrections and suggestions.

**Part II – Major Issues: Key Experiments Required for Acceptance**

Reviewer #1: I have no major issues, presented data justifies author conclusions.

Reviewer #2: In the abstract the authors say that this is the first time since 2022 that boost titers were higher against the homologous target than the ancestor, but throughout the paper they say this is the first time that this has ever happened. Which is it?

Fig 1A. The figure twice incorrectly says N478D instead of N487D. The figure neglects the addition of F456L in the XDV lineage. The figure includes LP.8.1.9 for no apparent reason. It was a minor lineage and not included in the experiments, I would remove it if there is no reason for it.

For recombinants XDV and XEC the Spike was derived from the parent shown. However, XFG got half of its RBD from LF.7 and the other half from LP.8.1.2. This is why the booster worked so well against XFG (which might be worth mentioning). The way the figure is laid out is inappropriate because it makes it look like 445R, 487D and 493E were acquired through mutation rather than recombination. I would recommend rearranging the figure to show LF.7 and LP.8.1.1 contributing to XFG, perhaps indicating that 445-493 was derived from LP.8.1.1.

**Part III – Minor Issues: Editorial and Data Presentation Modifications**

Reviewer #1: (No Response)

Reviewer #2: Given the high degree of convergent evolution in these lineages, I would recommend making a supplemental figure indicating what residue each lineage has at the heterogeneous positions, at least for the RBD. The data makes a lot more sense when you see it this way rather than simply based on ancestry.

I think it is worth stating explicitly that the ancestral lineage had the highest titer pre-boost, but LP.8 was the highest post-boost. It also think it’s probably also worth stating explicitly that that BA.3.2 had by far the lowest titer post-boost.

PLOS authors have the option to publish the peer review history of their article (what does this mean?). If published, this will include your full peer review and any attached files.

Reviewer #1: No

Reviewer #2: **Yes:**Marc Johnson

**Figure resubmission:**
---

## [Editor Report · Decision Letter 1]

30 Apr 2026

Dear Dr. Ho,

We are pleased to inform you that your manuscript 'LP.8.1-directed COVID-19 mRNA vaccines durably boost neutralizing antibodies and mitigate ancestral immune imprinting' has been provisionally accepted for publication in PLOS Pathogens.

Best regards,

Jesse D Bloom, Ph.D.

Guest Editor

PLOS Pathogens

Alexander Gorbalenya

Section Editor

PLOS Pathogens

Sumita Bhaduri-McIntosh

Editor-in-Chief

PLOS Pathogens

orcid.org/0000-0003-2946-9497

Michael Malim

Editor-in-Chief

PLOS Pathogens

orcid.org/0000-0002-7699-2064

The authors have addressed the minor comments, this paper is a valuable contribution.
---

## [Editor Report · Acceptance letter]

Dear Dr. Ho,

We are delighted to inform you that your manuscript, "LP.8.1-directed COVID-19 mRNA vaccines durably boost neutralizing antibodies and mitigate ancestral immune imprinting," has been formally accepted for publication in PLOS Pathogens.

Best regards,

Sumita Bhaduri-McIntosh

Editor-in-Chief

PLOS Pathogens

orcid.org/0000-0003-2946-9497

Michael Malim

Editor-in-Chief

PLOS Pathogens

orcid.org/0000-0002-7699-2064